# Contacting of Bicomponent TPU-Fibers with a Conductive Core: A Method for Data Acquisition and Analysis of the Electrical Properties

**Jeanette Ortega \*** , **Felix Krooß** , **Yuwei Stefan Li and Thomas Gries**

Institut für Textiltechnik, RWTH Aachen University, Otto-Blumenthal-Straße 1, 52074 Aachen, Germany
* Correspondence: jeanette.ortega@ita.rwth-aachen.de

**Abstract:** With the megatrend of digitalization, the demand for sensors in previously difficult-to-access scenarios is increasing. Filament-shaped sensors (FSS) are ideal for this demand, especially in applications in which the monitoring of textile structures is the focus. Electrically conductive bicomponent filaments based on thermoplastic polyurethane (TPU) and doped with carbon nanotubes (CNTs) offer great potential due to their flexible mechanical properties. Through the core-conducting, bicomponent structure, the sensing material is protected from environmental factors such as surrounding conductive materials and external moisture. The insulating material, however, simultaneously complicates the contacting method in order to measure sensing changes in the conductive core. In this work, laser cutting is employed as a technology in order to expose the conductive core of the filaments. The filament is then coated with silver and mechanically crimped, providing both a conductive interface for the data acquisition device as well as a protective layer. Laser parameters (power 20–100 W and speed 5–50 mm/s) are investigated to identify the parameters with the best cutting properties for which the filaments are analyzed visually and electrically. This work provides a robust and reproducible method for contacting core-conducting TPU filaments for strain-sensing applications. This study shows that while the choice of laser parameter influences the morphology of the cut surface, its impact on the resulting linear resistivity is negligible.

**Keywords:** filament-shaped sensor; TPU; CNT; laser cutting; bicomponent filament; melt spinning

## 1. Introduction

In the digital age, marked by a drive towards data-driven processes, the megatrends of Industry 4.0 and the Internet of Things (IoT) are increasingly coming into focus and bring with them the need for data acquisition. Specifically, in the manufacturing sector, a five-fold increase in data volumes is forecasted, creating a demand for sensor technologies in complex environments [1]. In this context, filament-shaped sensors (FSS) are gaining crucial importance due to their small size and flexibility. The filament form is especially well suited for integration into textile-based applications. The monitoring of fiber-reinforced composites has been of focus for many years, in which commercial strain gauges are applied to the external surfaces or fiber optical sensors (FOS) are integrated during the production [2,3]. Although these are important steps to the long-term structural health monitoring of such lightweight components, strain gauges can only measure at the point of application and FOS are often accompanied by expensive and complex measurement equipment. Additionally, the monitoring of civil structures through the sensor integration in geotextiles has been of interest [4]. Here, sensors provide early detection for needed maintenance but also alert locals about possible imminent dangers, such as landslides and sinkholes [5]. Lastly, filament sensors can be integrated into wearable textiles in order to monitor vital signals, such as breathing, which can provide early indications of respiratory problems in epileptic patients, therefore reducing the chance of sudden unexpected death in epilepsy (SUDEP) [6]. Seeing as how these applications have a wide range of mechanical

properties as well as production processes, there is a demand for versatile FSS, which provide quantitative information about the current strain in the component or structure.

In order to cover large strain ranges, an elastomeric material is to be chosen. With a refinement to a thermoplastic elastomeric material, this material can be processed into filaments through the industrialized melt-spinning process. Through the doping of this elastomeric material with conductive nanoparticles, an electrical conductivity can be achieved, which is utilized for the resistive-based sensor principle. In order to fulfill these technical requirements, a nanocomposite material consisting of thermoplastic polyurethane (TPU) and carbon nanotubes (CNTs) is chosen as the conductive, sensing material.

There is no shortage of work reporting the development of nanocomposites of TPU and CNTs. Much of the research has focused on understanding the influence of the nanoparticle concentration on the resulting static resistivities. A percolation threshold has been reported, below which no conductive network is formed in the nanocomposite and above which additional doping of the nanoparticles does not contribute to an improvement of the electrical properties. These works have shown that a CNT concentration of 4 wt.% in TPU is generally sufficient in order to employ the filament as a sensor [7–9].

Despite the detailed previous work, the filaments are always produced in a fashion so that the filament surface is the electrically conductive component [7–9]. This may be beneficial if the sensor is meant to detect moisture or electrical signals, but detrimental when strain is to be sensed in a possible environment of moisture, such as rain or sweat, or in the vicinity of other conductive components, such as carbon fiber. In this work, a core conducting, bicomponent filament with an insulating sheath (schematically shown in Figure 1) is developed as well as a contacting method for the electrical analysis of such filaments.

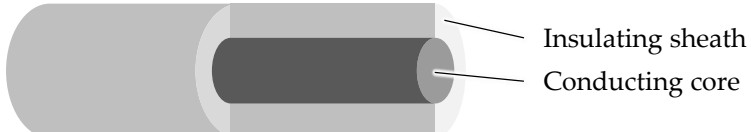

Insulating sheath
Conducting core

**Figure 1.** Schematic representation of the developed core conducting, bicomponent filament.

## 2. Materials and Methods

### 2.1. Materials

#### 2.1.1. Sheath Component

An insulating sheath is coextruded around the conductive core, which serves to protect the conductive core from the environment. The selected TPU sheath allows for high strains and elasticity during cyclic loading. The material is the commercially available Elastollan® 1185 from BASF Polyurethanes GmbH, Lemförde, Germany. It will be further referred to as TPU1185.

#### 2.1.2. Core Component

A nanocomposite material is a multiphase material, in which at least 1 of the phases measures less than 100 nm in 1 of the dimensions. In this work, multi-walled carbon nanotubes (CNTs) are compounded together with a thermoplastic polymer in order to achieve a conductive polymer nanocomposite. To further ensure the high stretchability and elasticity of the filaments as well as a good adhesion between the core and sheath components, a TPU is chosen as the matrix material for the core conducting nanocomposite.

It is widely known that the properties of the starting compound are critical for the properties of the final product. The dispersion and agglomeration of the compounded particles can rarely be improved upon after the initial compounding process. For this reason, the compound employed in this work is a commercially available material, for which the compounding process has been optimized and a homogeneous product can be produced. The material in this work is a thermoplastic polyurethane (TPU) base with 4 wt.% CNTs by

the name PLASTICYL$^{TM}$ TPU from the company NanoCyl SA, Sambreville, Belgium. This material will be further referred to as TPU0401.

*2.2. Processing Methods*

2.2.1. Filament Production

Due to the hydroscopic nature of TPU, the granulate materials are dried previously to extrusion, establishing a residual moisture content of <200 ppm [10]. For this purpose, the oven LUXOR A from the company motan holding gmbh, Constance, Germany, is used. The materials are dried at 80 °C with hot air for 16 h. The moisture contents are measured by Karl–Fischer titration using the Coulometric KF Titrator C30 from the company Mettler Toledo, Inc., Columbus, OH, USA. The moisture content was reduced from 794.8 ppm (+/−35.16 ppm) to 114.23 pmm (+/−9.08 ppm) and 1115.73 ppm (+/−49.17 ppm) to 126.10 ppm (+/−18.73 ppm) for the TPU 1185 and TPU0401, respectively.

The core and sheath materials are formed into filaments using the bicomponent, monofilament melt-spinning process. In melt spinning, thermoplastic polymers are fed from a hopper to an extruder, in which a rotating screw is employed along with multiple heating elements to melt and liquify the polymer. Through the heating elements as well as the resulting shear stresses, heat is incorporated into the polymer. This molten polymer is then fed further to a gear spin pump, in order to ensure a constant volume throughput. After the spin pump, filters in the spinning pack are employed in order to remove foreign material and/or gelled polymer. Finally, the polymer passes through the spinneret with a defined hole diameter and capillary length from which the filament receives its form. For the case of bicomponent spinning, two hoppers, extruders, and spin pumps are employed for a second polymer stream. These two polymer streams are then fed together into the spinning pack, consisting of the filters and spinneret, where they remain separated until joining in the capillary of the spinneret. Schematic representations of the melt-spinning machine as well as the spinneret have been previously shown in [11]. The precise spinning parameters are shown in Table 1.

**Table 1.** Spinning parameters for the investigated filament.

| Parameter | | Unit | Value | |
|---|---|---|---|---|
| | | | **Core** | **Sheath** |
| Extruder pressure | | bar | 30 | 30 |
| Extrusion temperature | Zone 1 | °C | 177.5 | 175.0 |
| | Zone 2 | | 187.5 | 185.0 |
| | Zone 3 | | 192.5 | 200.0 |
| | End piece | | 197.5 | 215.0 |
| | Melt line | | 202.5 | 220.0 |
| | Pump block | | 225.0 | 220.0 |
| | Spinneret | | 225.0 | 220.0 |
| Pump size | | cm$^3$/min | 0.3 | 0.6 |
| Pump speed | | RPM | 6 | 20 |
| Spinneret capillary hole diameter | | mm | 1.1 | |
| Spinneret capillary hole length | | mm | 2.2 | |
| Godet speed 1 | | | 28 | |
| Godet speed 2 | | m/min | 29 | |
| Winding speed | | | 30 | |

2.2.2. Filament Cutting and Contacting

In order to analyze the electrical properties, a contact must be made with the core component of the filament. A standard contacting method for metallic wires is to strip the wire of the insulating plastic layer. Such a method is not suitable for the contacting of the bicomponent filament because of the strong interfacial adhesion between the core and sheath components. As an alternative to stripping the insulation layer, the conductive surface can be increased through the employment of silver paint. For this, the bicomponent

filament is cleanly cut in the cross-section and the cut surface is dipped in silver paint. To protect the silver layer from mechanical damage, such as bending or flaking, a metal crimp is finally added to the surface of the silver paint. The conductive interfaces generated by this method are shown in Figure 2.

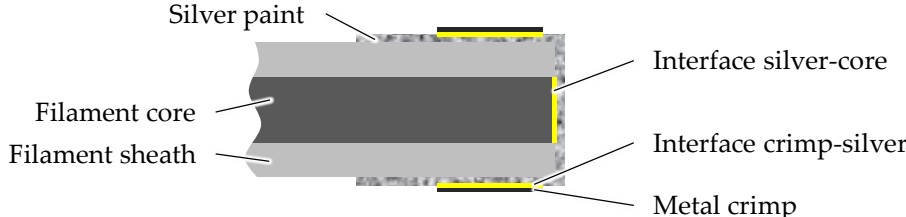

**Figure 2.** Schematic of the longitudinal filament cross-section showing the electrical interfaces after contact.

Three cutting methods are tested in this work: (1) scalpel at room temperature, (2) scalpel after freezing the filament, and (3) laser at room temperature. A total of five samples are prepared for further analysis by each of the three cutting methods. The laser employed in this work is the R500 Rayjet from the company Trotec Laser GmbH, Marchtrenk, Austria. This laser is generally used for cutting or engraving various materials, including wood, paper, leather, and plastics. The main features of the laser cutting machine from this work are shown in Table 2.

**Table 2.** Features of the laser cutting machine R500 Rayjet [12].

| Parameter | Unit | Value |
|---|---|---|
| Working area | mm × mm | 1300 × 900 |
| Laser source | - | $CO_2$ |
| Laser wavelength | nm | 655 |
| Laser class | - | 2 |
| Max. cutting speed | m/s | 0.5 |
| Max. laser power | W | 100 |
| Max. laser frequency | Hz | 1000 |
| Software | - | Rayjet Manager V11.3 |

After the core of the filament is exposed through cutting, it is dipped 4 mm in silver paint, covering the cut surface as well as the insulating sheath material. The silver paint used in this work is from the company MARAWE GmbH & Co. KG, Regensburg, Germany. A microscopic image of a cut and silver-painted filament can be seen in Figure 3.

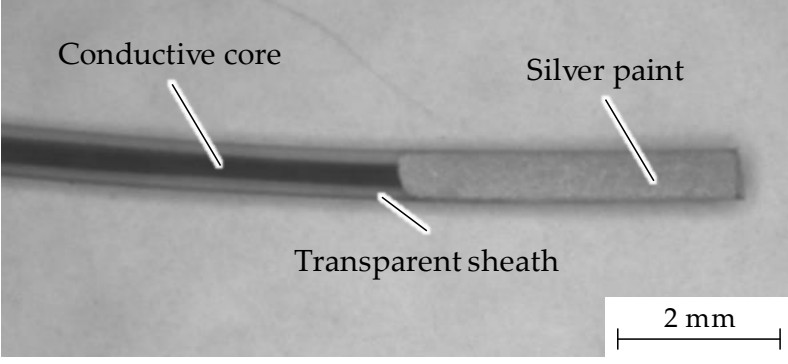

**Figure 3.** Image of a cut and silver painted bicomponent filament.

*2.3. Analysis Methods*

2.3.1. Scanning Electron Microscopy

In order to qualitatively analyze the cut surface, scanning electron microscopy (SEM) is employed. The sample preparation consists of sputtering the as-cut (razor or laser) surface with gold. The SEM used in this work is the FlexSEM 1000 II from the company Hitachi High-Technologies Corporation, Tokyo, Japan.

2.3.2. Static Electrical Resistivity

In addition to the qualitative SEM images, the static electrical resistivity of the filaments is measured in order to determine if the varied cutting parameters have a significant influence on the measured resistivity. After successfully cutting and contacting, the resistance is measured with the multimeter Series 2100: 6 ½ Digit from the company Keithley Instruments, Cleveland, OH, USA, in DC current. A four-point measurement method is used. In order to evaluate the contact resistances of the multiple interfaces, samples are cut to 3 different lengths (10, 15, and 20 cm) and the resulting resistance is plotted in relation to the cut length. Ideally, the data points form a linear trend and the y-intercept of the extrapolated trendline goes to 0. The linear resistance is calculated using the equation below:

$$R_L = \frac{R}{L},$$ 
(1)

where R is the resistance in MΩ, L is the length of the filament in m, and $R_L$ is the linear resistivity in MΩ/m.

**3. Results**

The results of this work are separated into two sections. First, the pretrials of the cutting with the razor at room temperature after freezing and laser cutting will be described. Subsequently, the determination of the optical laser cutting parameters based on qualitative and quantitative analysis will be presented. For all the trials, the filament described in Table 1 is used.

*3.1. Cutting Pretrials*

In the preliminary experiments, fibers are cut with scalpels. The direction of the cut is perpendicular to the fiber axis. A noticeable yield of the soft material is observed during cutting. To prevent crushing of the cross-sectional area, additional fibers are stored in the freezer at −8 °C for at least 24 h before cutting. Further attempts to cut the fibers longitudinally were abandoned due to the absence of a manual, reproducible method.

For each of the cutting methods, 5 samples are imaged with the SEM at 2 magnification levels (100× and 300×). The core and sheath of the filaments are visible due to a slight difference in the grey tones of the higher-magnified images. For the preliminary trials, the laser power is set to 60 W and the laser speed is set to 25 mm/s. Three representative images per cutting method are shown in Figure 4.

In addition to the qualitative measurements, the filaments are contacted with silver paint and metal crimps, and the electrical resistivities are measured. Unfortunately, the samples cut with the razor, at room temperature and frozen, are unable to be measured because the maximum resistance of the multimeter (100 MΩ) at all lengths is reached and only "Overload" is shown on the display [13]. The resulting linear resistivity of the laser-cut sample could be measured and is 68.21 (+/−9.87) MΩ/m.

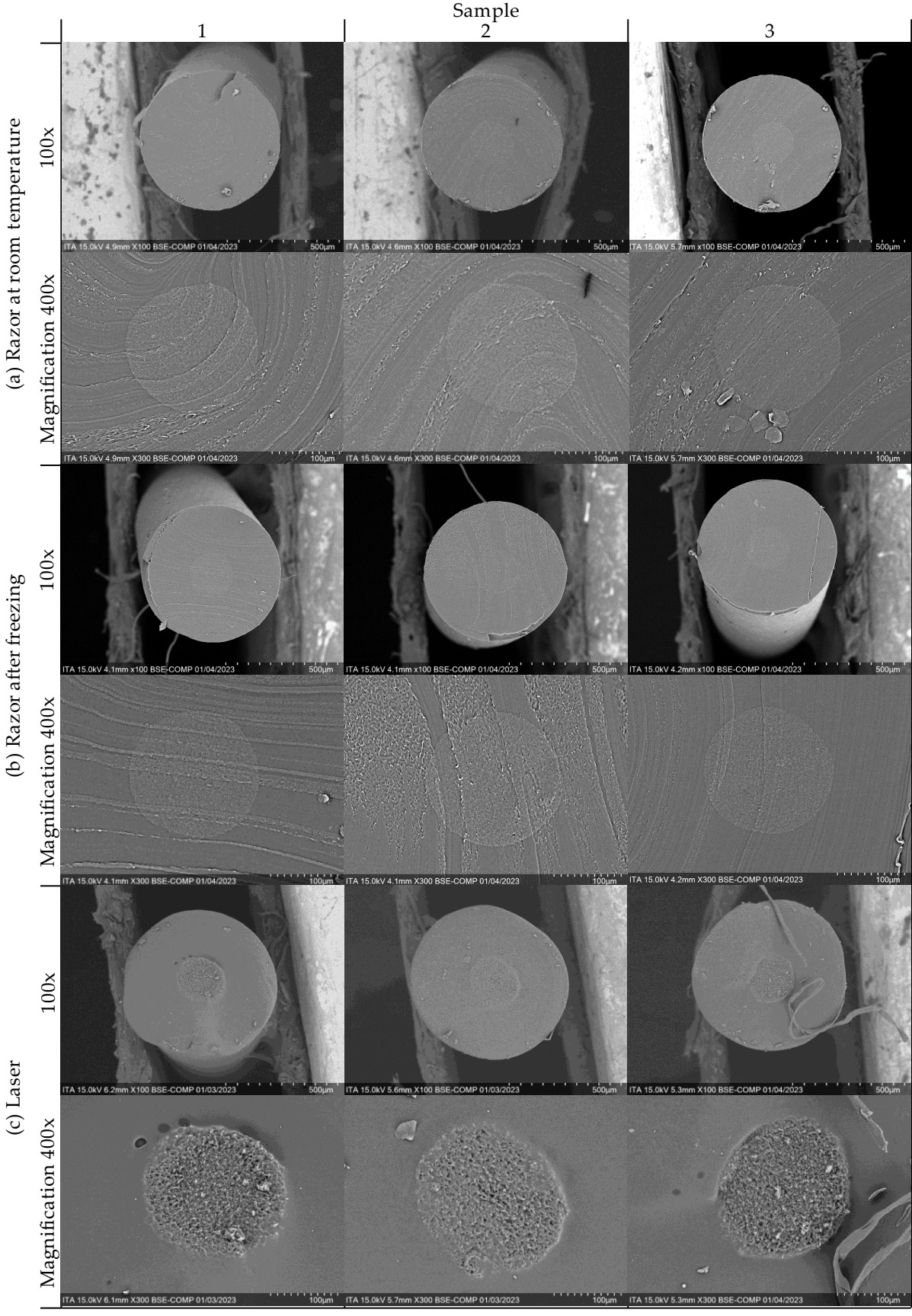

**Figure 4.** Cross-sectional SEM images of the three cutting methods; (**a**) razor at room temperature, (**b**) razor after freezing, and (**c**) laser.

### 3.2. Determination of the Optimal Laser Parameters

After evaluation of the cutting pretrials, the parameters of the laser cutting machine are investigated in order to determine if they have an influence on the quality of the cut surface and, therefore, on the electrical contact. For this purpose, the laser speed and laser power are varied. A $2^2$ fully factorial design of experiments (DOE) was planned, but the combination of the highest speed with the lowest power (50 mm/s and 20 W) was not able to cut through the filament. This is thought to be due to the fact that enough energy cannot be transferred to the filament in the short contact time. The resulting parameters tested are shown in Table 3. As a consequence, 2 $2^2$ DOEs are analyzed: (1) 5 vs. 25 mm/s and 20 vs. 100 W, and (2) 5 vs. 50 mm/s and 60 vs. 100 W. The qualitative results from the SEM images are shown in Figures 5 and 6.

**Table 3.** Varied laser-cutting parameters. (✓ parameter tested, **x** parameter not tested).

|  |  | Laser Power (W) | | |
|---|---|---|---|---|
|  |  | **20** | **60** | **100** |
| Laser speed (mm/s) | 5 | ✓ | ✓ | ✓ |
|  | 25 | ✓ | ✓ | ✓ |
|  | 50 | **x** | ✓ | ✓ |

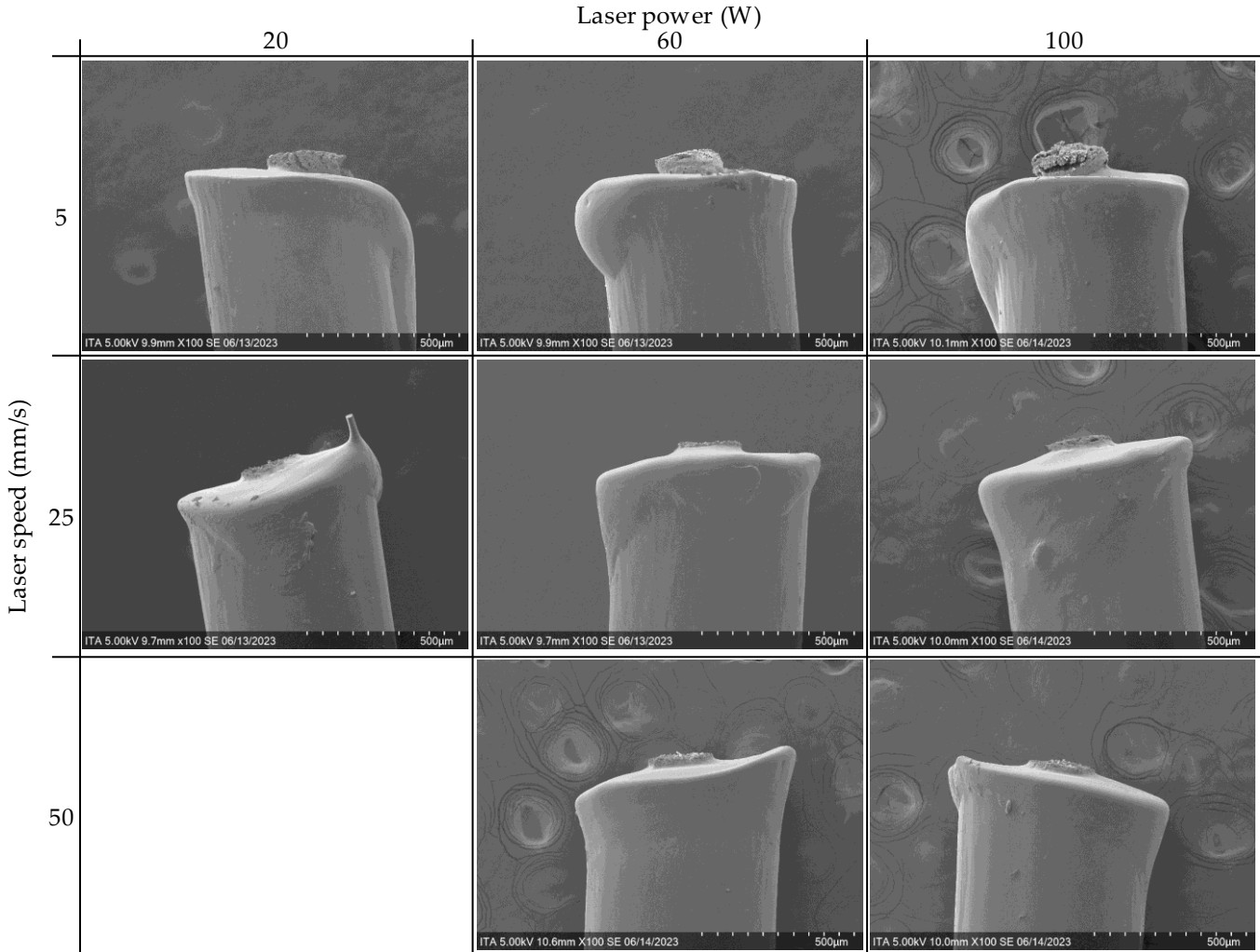

**Figure 5.** Longitudinal SEM images of comparison of the laser parameters.

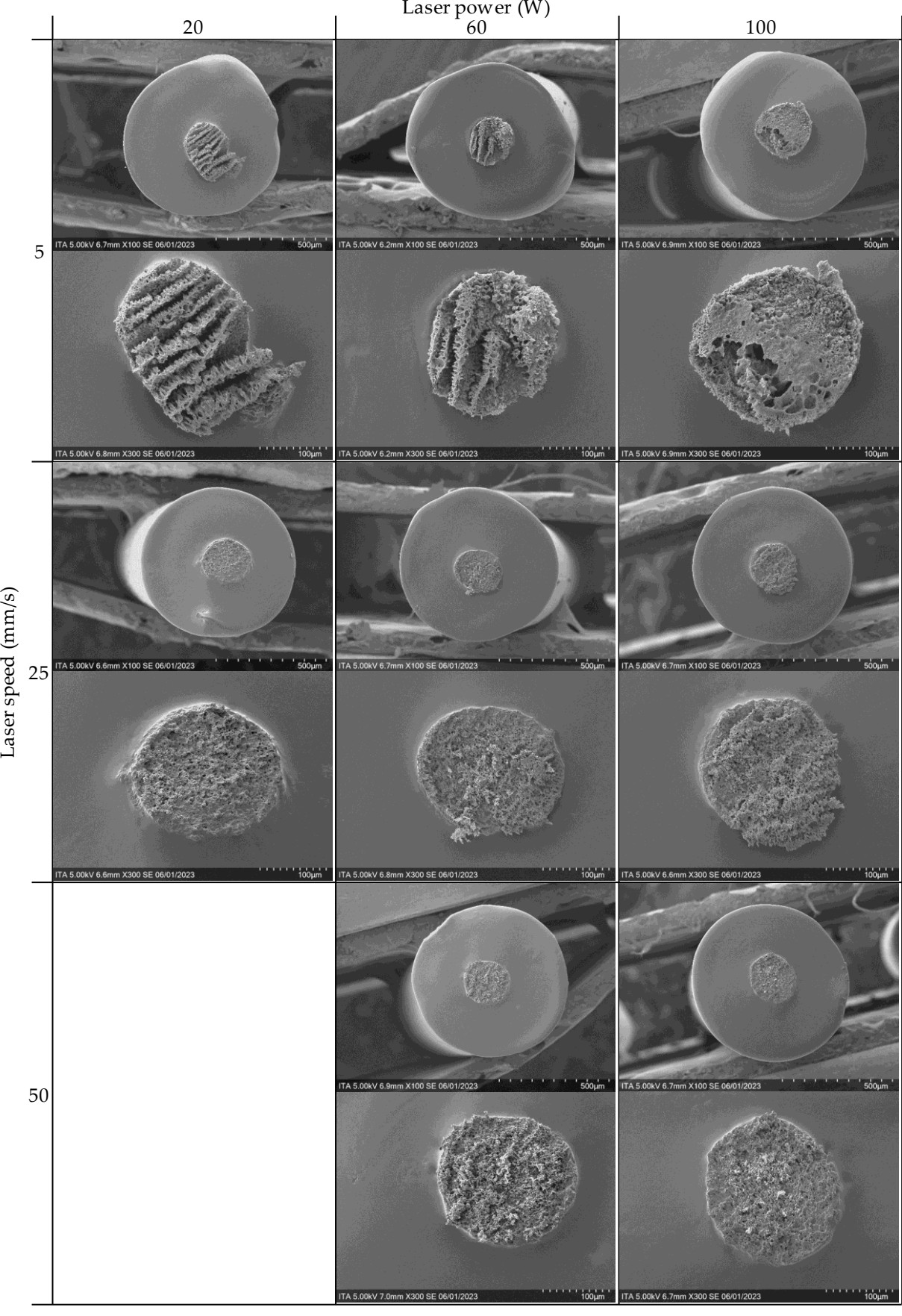

**Figure 6.** Cross-sectional SEM images of comparison of the laser parameters.

The samples are also tested regarding the static linear resistivity. The best cutting parameters are those with the following characteristics:

- Low average linear resistivity;
- Low standard deviation of the linear resistivity;
- Low absolute value of the y-intercept (length vs. resistance);
- Linearity $R^2$ close to 1 (length vs. resistance).

The resulting data regarding the linear dependence of resistance are shown in Figure 7. The interaction plots of the laser power and laser speed on the linear resistivity, y-intercept, and linearity $R^2$ can be seen in Figures 8–10 for both investigated DOEs, as described in Section 3.2. Furthermore, the Pareto charts of the standardized effects of the linear resistivity and the laser parameters (A: laser speed, B: laser power) can be seen in Figure 11. For this statistical analysis, a level of confidence of 95 % ($\alpha = 0.05$) is taken. Depending on the data and the number of samples tested (n = 15), a minimum effect, the reference line, is calculated, and above which, the factor is considered significant. For Comparisons 1 and 2 in Figure 11, these are 10.51 and 5.05, respectively.

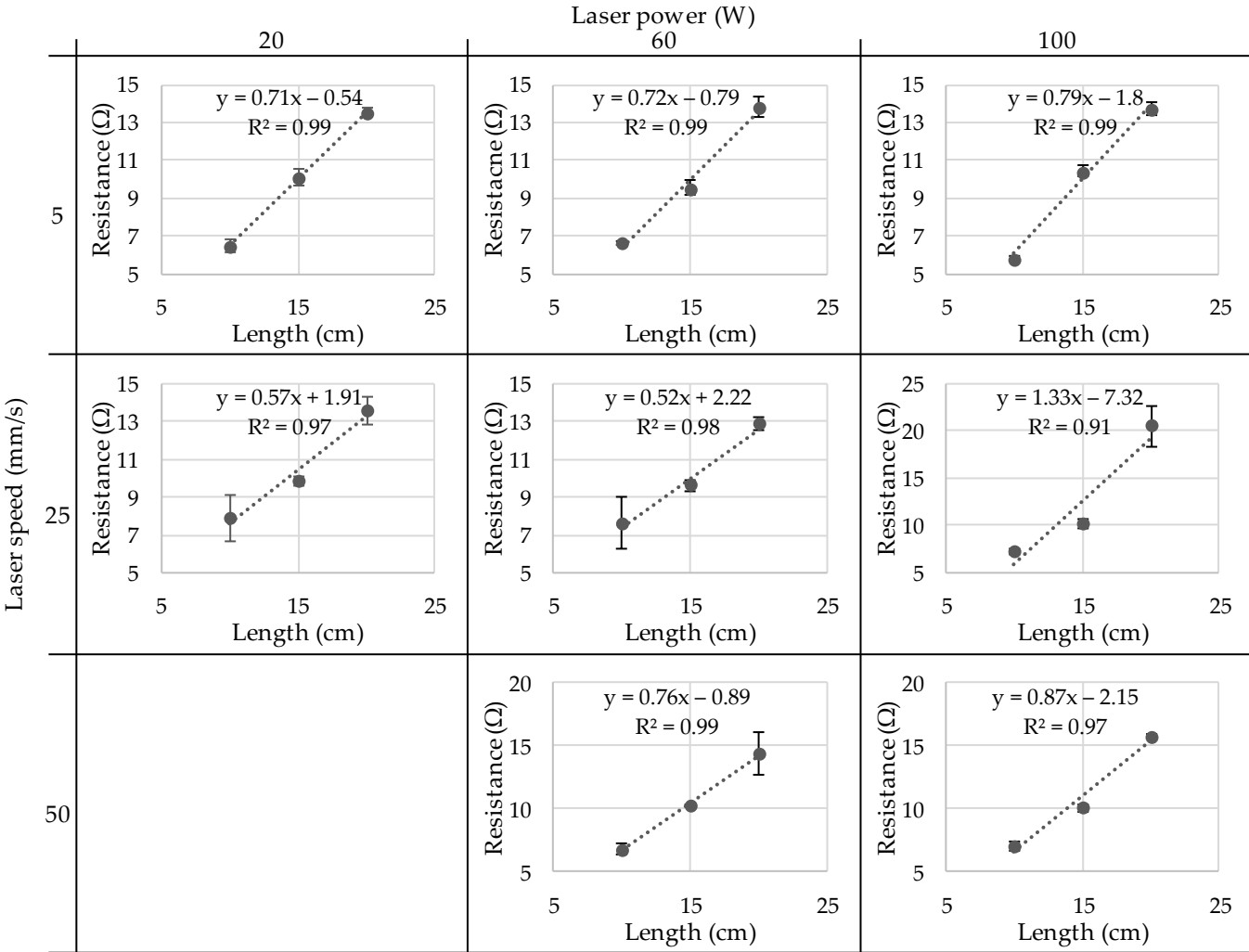

**Figure 7.** Linear dependence of resistance and sample length.

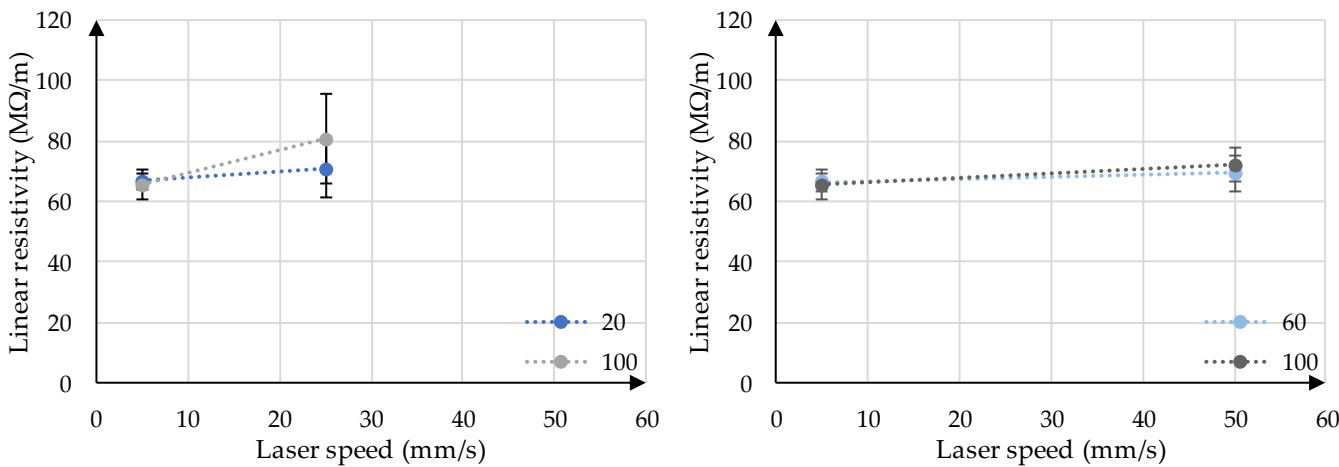

**Figure 8.** Interaction plot linear resistivity and laser parameters.

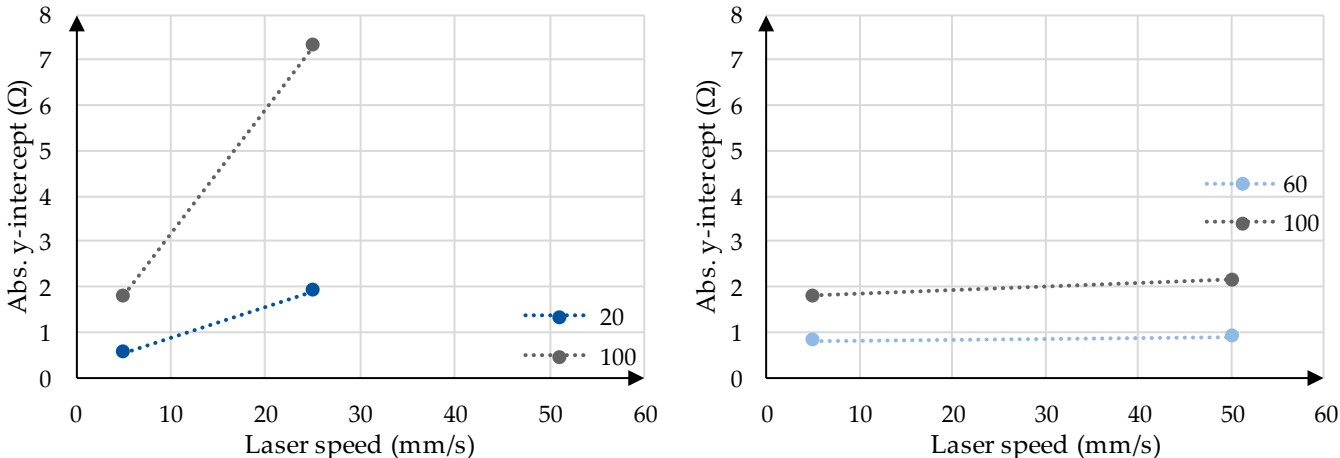

**Figure 9.** Interaction plot abs. y-intercept and laser parameters.

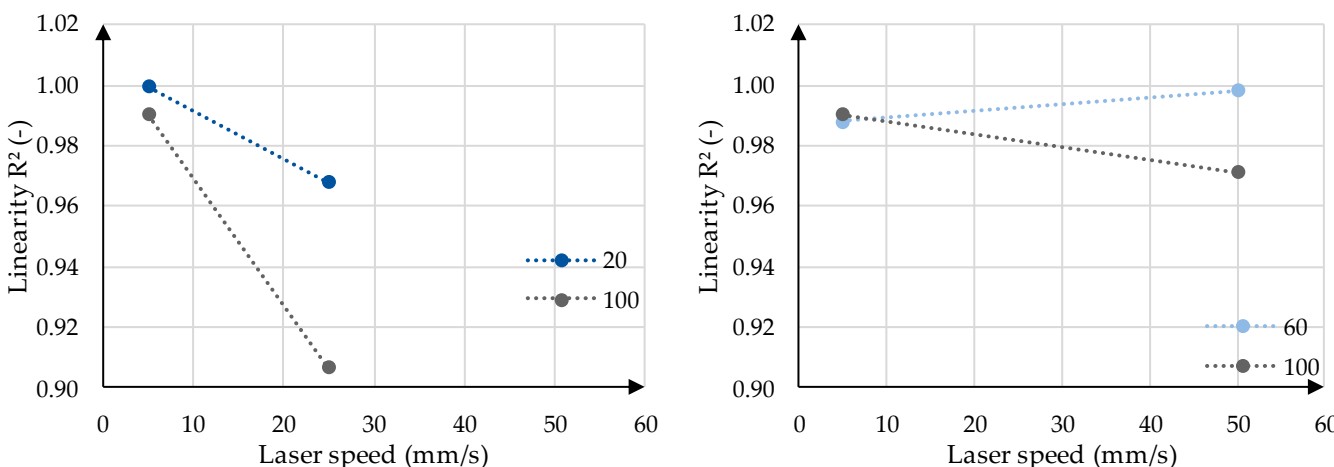

**Figure 10.** Interaction plot linearity $R^2$ and laser parameters.

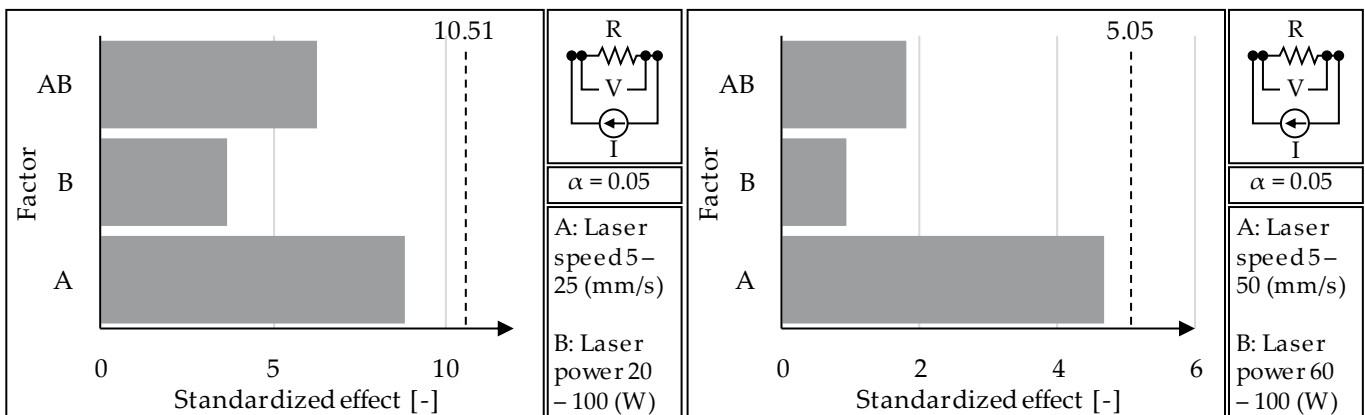

**Figure 11.** Pareto charts of standardized effects of linear resistivity and laser parameters.

## 4. Discussion

In this chapter, the presented results will be discussed with explanations for phenomena observed. First, the SEM images of the pretrials will be evaluated. Subsequently, the SEM images as well as the static linear resistivities of the parameter optimization are discussed.

### 4.1. Cutting Pretrials

In the SEM images, no clear difference can be observed when comparing the razor-cut samples cut at room temperature and after freezing. On the other hand, there is a clear difference in the surface properties of the razor and laser-cut samples, especially visible in the highly magnified (300×) images. This comparison can be seen in Figure 12.

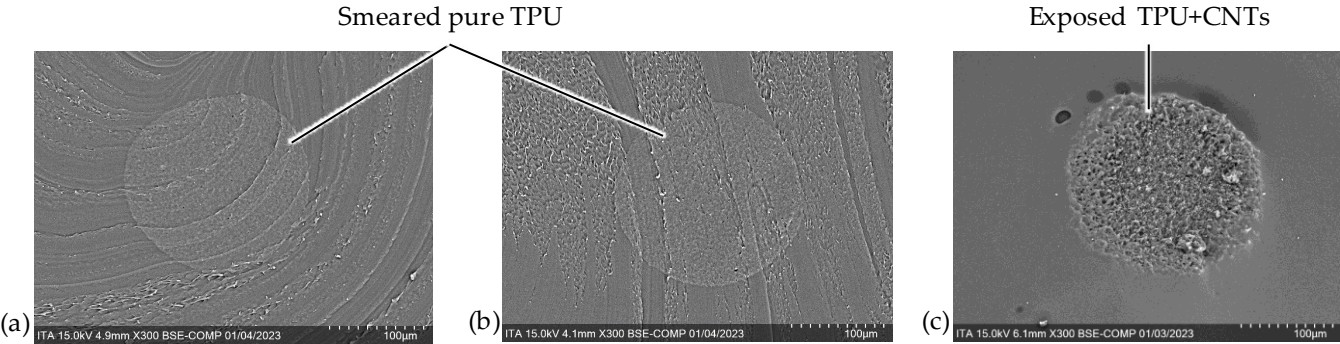

**Figure 12.** Direct comparison of razor and laser-cut samples; (**a**) razor at room temperature, (**b**) razor after freezing, and (**c**) laser.

It appears that due to the cutting with the razor, the pure TPU from the sheath component is smeared over the conducting core. This would explain the inability to measure the resistance with the multimeter, as the sample, not allowing contact with the conductive core, essentially has an infinite resistance. This is a phenomenon that is not observed when razor-cutting filaments with a different sheath material, such as polypropylene (PP) [11]. This is thought to be due to the different mechanical properties of the sheath materials. TPU is a very soft and elastic material, even at room temperature. Conversely, PP is rather stiff and brittle. This difference makes the cutting and contacting of bicomponent filaments with PP in the sheath relatively trivial, whereas TPU poses a new problem.

Cutting the filaments with the laser essentially burns away the polymer material at the cut surface. This thermal rather than mechanical removal of the material allows for a cleaner cut of the surface. Furthermore, there is a clear difference in the surface of the

conducting core and the insulating sheath for the laser-cut samples. It is assumed that the CNTs remain in the core to a certain extent and this network structure is then visible as small cavities. This phenomenon is discussed in more detail in the following chapter.

### 4.2. Determination of the Optimal Laser Parameters

4.2.1. Scanning Electron Microscopy

When analyzing the cross-sectional SEM images of Figures 5 and 6 some qualitative characteristics can be observed. In only evaluating the 3 samples cut at 5 mm/s, all have larger and deeper cavities in the core component in comparison to those cut at the higher laser speeds of 25 and 50 mm/s. This is thought to result from the relatively long dwell times of the laser while cutting. Additionally, among the 3 samples cut at 5 mm/s, a difference can be seen from the lower to the higher laser power. At lower powers, the core has a striped visual effect, whereas the core cut at the higher powers has a more irregular pattern and rather larger, rounder cavities (Figure 13). It is thought that the regular striped pattern at the lower laser powers results from the pulsing of the laser while moving and cutting the material. These cavities may represent the removal of the CNTs, TPU, or both. On the other hand, at higher power (5 mm/s and 100 W), the amount of energy input is so high and the dwell time is so long (due to the slow speed) that the delicate structures remaining at the lower powers are destroyed, leaving larger vacancies in the core. This hypothesis would need to be further investigated in future research.

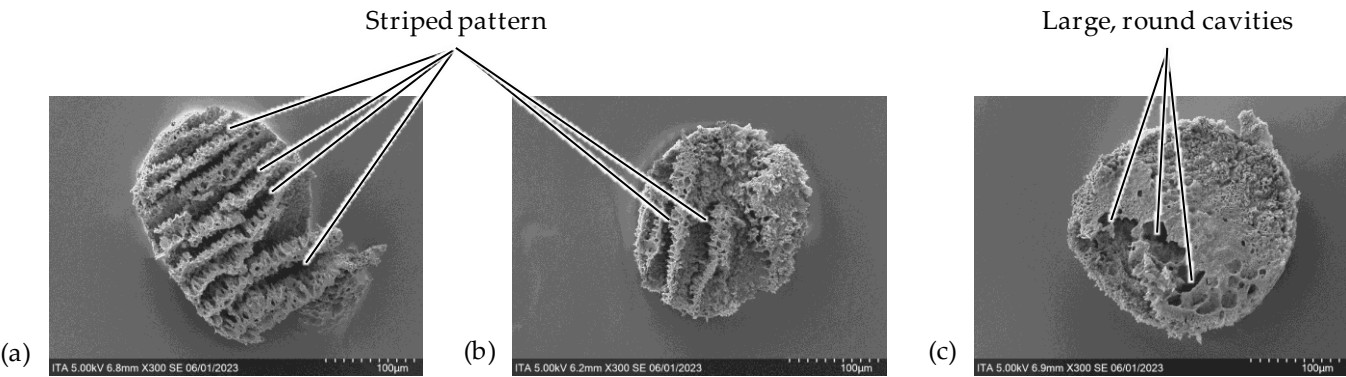

**Figure 13.** Comparison of surface pattern with varied laser power at 5 mm/s; (**a**) 20 W, (**b**) 60 W, and (**c**) 100 W.

In comparing the samples cut at higher speeds, the SEM images appear relatively similar; at higher speeds, the laser power has less of an influence on the surface. However, the resulting cavities of the samples cut with the 2 lowest powers (25 mm/s and 20 W; 50 mm/s and 60 W) appear deeper as stronger shadows in the core material. This fine and deep structure may mean that more of the TPU is burned away while cutting leaving the intricate CNT network visible. Conversely, at the higher powers, more of the complete material, not just TPU, is removed, leaving a more homogeneous surface. These are preliminary assumptions and would need to be further analyzed.

Lastly to be investigated from the cross-sectional images is the small tail visible in the samples cut at 25 mm/s and 20 W, which is also visible in the longitudinal images (Figure 14). This may result from the relatively fast but low-energy input of the laser. There is not enough time for the sheath material to be fully and cleanly cut and the remaining "tail" is broken after cutting. This would also explain why cutting with even higher speed was not possible at the lowest power (50 mm/s and 20 W). The even higher speed and less dwell time for cutting resulted in an uncut filament, which was macroscopically visible.

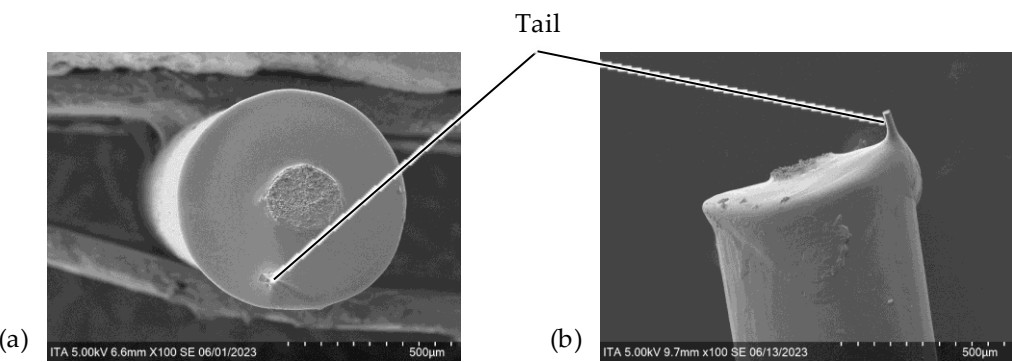

**Figure 14.** Tail resulting from cutting at high speeds and low power (25 mm/s and 20 W); (**a**) cross-sectional and (**b**) longitudinal SEM images.

With analysis of the longitudinal images, other conclusions can be made. It can be seen that the cores of the samples cut at low speeds (5 mm/s) protrude more from the sheath than the samples cut at higher speeds (25 and 50 mm/s). This dependence of the protrusion on the laser parameters can be seen in Figure 15.

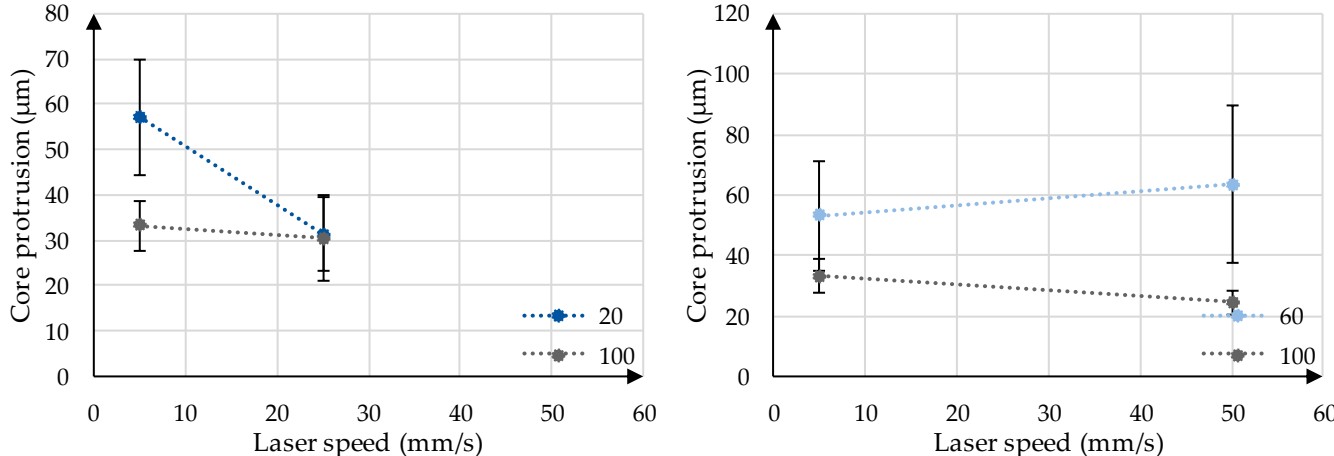

**Figure 15.** Interaction plot core protrusion and laser parameters.

This could be explained by different melting characteristics of the sheath material during cutting due to different thermal properties. It is thought that the cut height of the laser is in line with the residual core component and not in line with the remaining sheath. The material at the cut interface is burnt away and material near the interface melts due to residual heat. This melting causes the sheath to retract and collect around the solid sheath. This is shown schematically in Figure 16. This phenomenon can also be observed in the laser processing of carbon fiber-reinforced plastics (CFRPs), where there is a heat-affected zone of carbon fiber and polymer matrix [14].

This thicker sheath near the cutting interface can be observed to different degrees for all the samples in Figure 5. For the slower speed of 5 mm/s, however, the collection of the sheath is more prominent and, in some cases, appears to have even "dripped" down the filament. Additionally, the collection of the material seems to be more prevalent on one side of the filament. For the higher speeds, the thickening is less extreme and results in more of a diagonal cut as for the lower speed. It is hypothesized that the side of the large collection for the slow speed and the higher diagonal side for the higher speed is the bottom of the filament while cutting; the filaments are laid horizontally in the machine. Due to gravity, the melted sheath material may flow more to the lower side of the filament before resolidifying, either as the round collection or the diagonal cut. This could be further

investigated by imaging filaments not fully cut or by maintaining the filament position between cutting and imaging.

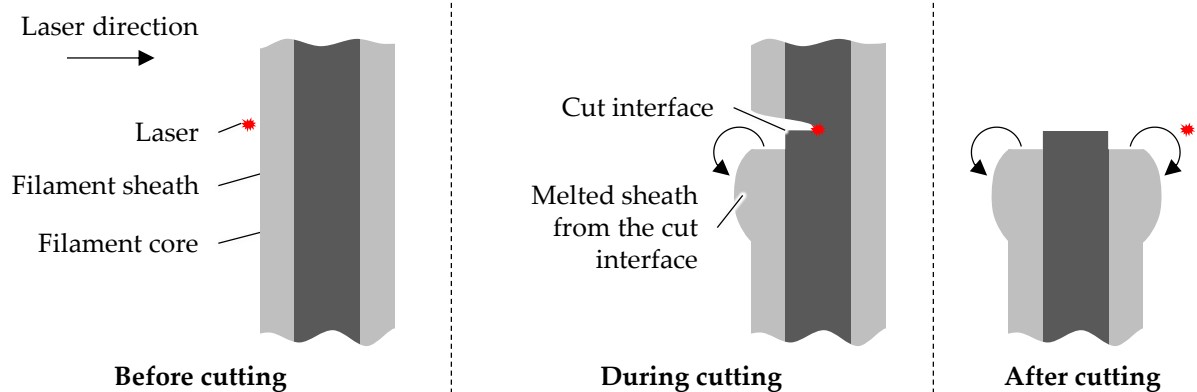

**Figure 16.** Schematic representation of the laser-cutting process.

### 4.2.2. Static Electrical Resistivity

In Figure 7, it is observable that all the samples tested are relatively linear and that the data, when extrapolated to 0 cm, also tend towards 0 Ω. This is important because a non-zero y-intercept points towards either a contact resistance between the interfaces or inhomogeneous filaments and contacting method. This can be seen in Figures 9 and 10 for the y-intercept as well as the linearity $R^2$. It can be generally said that both a lower laser speed and lower laser power lead to a smaller y-intercept. The lower speed and power additionally lead to a higher linearity, with the exception of 5 mm/s and 60 W, although the $R^2$ of 0.9881 is already very close to the ideal 1.0. Because these values, y-intercept and linearity, are calculated using all data points per sample, there is no possibility to calculate the significance with a *t*-test, and, therefore, only the trend of the data, but not the significance of the data, can be concluded.

Conversely, the static linear resistivity can be evaluated for both the trends and the significance of the parameter variation. Again, the trend shows that the lower laser speed and lower laser power tend towards a lower static linear resistivity average and standard deviation, whereby the difference appears more dramatic for the standard deviation. The significance of the static linear resistivity is shown in Figure 11 with the help of a Pareto chart. In both comparison cases, both factors, laser speed (A) and laser power (B), as well as the combination of the parameters (AB) are found to be not significant. For this reason, based solely on the resulting linear resistivity, there is no evidence to select certain laser parameters. However, based on the observed trends, keeping in mind that the difference may not be statistically significant, it is suggested to conduct further cutting and contacting of bicomponent TPU filaments with the 5 mm/s laser speed and 20 W laser power.

### 5. Limitations

This work is intended to serve as a basis for making FSS with conductive core measurable. The scope is limited to TPU fibers with a CNT-doped core. Deviating fiber properties, such as material, thickness, and core-to-sheath ratio, were not examined and may lead to different ideal cutting parameters. The literature in the field of CFRP research suggests that the carbon in the CNTs undergoes a heat-induced morphological transformation to amorphous graphite. Whether and to what extent this is applicable to the fibers under investigation was not examined. The measurements are limited to the electrical properties.

### 6. Conclusions and Outlook

The motivation for this work is the trend towards digitalization in industrial applications. This requires novel sensor systems, including filament-formed sensors for applications in which textiles are applied, such as fiber-reinforced composites, geotextiles,

and smart wearables. Bicomponent filaments allow the sensitive conductive core material to be protected from external influences such as moisture and wear. TPU has the advantage that the material is very flexible and variable, able to cover a wide range of applications. However, the soft nature of TPU makes the contacting of the conductive core in a bicomponent filament challenging.

In this work, a cutting and contacting method for TPU core-conductive bicomponent filaments is developed. The cutting is carried out with a laser cutter in order to avoid smearing the sheath TPU material. With silver paint and metallic crimps, the conductive area is enlarged and simultaneously protected from friction and flaking. Differences in the surface morphology of the cut samples were observed while altering the laser parameters (laser speed and laser power). However, it was found that these parameters do not have a significant influence on the resulting linear resistivity of the samples. Despite this lack of significance, it is suggested to continue cutting the bicomponent filament with the lower laser speed (5 mm/s) and lower laser power (20 W) based on the trends of the average and standard deviation of the linear resistivity, as well as the y-intercept and linearity of the length vs. resistance trendlines.

This work serves as the starting point for much more research, as it makes the electrical analysis of TPU-bicomponent filament even possible through stable cutting and contacting. For further work, the depth of the heat transfer into the core will be evaluated with monocomponent filaments. Additionally, the thickening of the sheath material after cutting can be investigated. Lastly, countless more filaments, with various production parameters, such as core pump speed, extrusion temperature, spinneret diameter, and winding speed, can be examined. This will further the research in the development of filament-based strain sensors and will give information about the dependence of the static and dynamic electrical resistivities on the spinning parameters, possibly allowing for a tailored filament for each application case.

Summarized Conclusion:

- The cutting of core-conductive bicomponent TPU filaments using a laser cutter avoids smearing and allows contacting using silver paint and metallic crimps for enhanced durability.
- Laser parameters (speed and power) were found to not significantly affect the linear resistivity of the filaments, suggesting optimal cutting conditions at lower speed (5 mm/s) and power (20 W) based on the observed trends in resistivity and surface morphology.
- Further investigations into heat transfer, sheath material thickening post-cutting, and the effects of various production parameters on filament performance are proposed.
- Advancements in filament-based strain sensors and understanding the impact of spinning parameters on electrical resistivities open avenues for customized sensor applications in diverse industrial fields.

**Supplementary Materials:** The following supporting information can be downloaded at: https://www.mdpi.com/article/10.3390/fib12050041/s1.

**Author Contributions:** Conceptualization, J.O.; data curation, Y.S.L.; formal analysis, Y.S.L.; funding acquisition, T.G.; investigation, J.O., F.K. and Y.S.L.; supervision, J.O. and T.G.; writing—original draft, J.O.; writing—review and editing, J.O. and F.K. All authors have read and agreed to the published version of the manuscript.

**Funding:** This research was funded by the Federal Ministry for Economy and Energy (Bundesministerium für Wirtschaft und Energie, BMWi) under the program "Central Innovation Programme for SMEs" (Zentrales Innovationsprogramm Mittelstand, ZIM), grant number KK5055907ZG0.

**Data Availability Statement:** The data presented in the manuscript and Supplementary Materials are available upon request from the corresponding author.

**Acknowledgments:** The authors would like to thank BASF Polyurethanes GmbH, Lemförde, Germany, for providing the material TPU 1185 for this research.

**Conflicts of Interest:** The authors declare no conflicts of interest. The funders had no role in the design of the study; in the collection, analyses, or interpretation of data; in the writing of the manuscript; or in the decision to publish the results.

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
