# Peer review of "Contacting of Bicomponent TPU-Fibers with a Conductive Core: A Method for Data Acquisition and Analysis of the Electrical Properties"

_fibers, doi:10.3390/fib12050041_

Round 1

Reviewer 1 Report

Comments and Suggestions for Authors

In this work, authors fabricate a cutting and contacting method for TPU core-conductive bicomponent filaments. The cutting process is performed using a laser cutter to prevent any potential smearing of the TPU material sheath. By incorporating silver paint and metallic crimps, not only is the conductive area enlarged but it also receives enhanced protection against friction and flaking. In general, this work provides interesting findings. I recommend that Fibers could accept this work with the following amendments.

1. There is a lack of clarity in the correspondence between the text and image in paper.’

2. It is better for the authors to add a description to the cross-sectional SEM images (Figure 4) to explain the difference between these three cutting modes.

3. The author should provide a comprehensive description and thorough analysis of the findings presented in figure 5 and 6, rather than solely presenting them without further elaboration.

4. The author should consider citing the relevant literature to provide theoretical support for the analysis of the results presented in figure 15.

Comments on the Quality of English Language

Author Response

Dear Reviewer,

thank you for taking your time to review our paper. Please find attached our responses to your feedback.

  1. The coherence between text and images in the paper has been revised and modifications have been made, in order to improve the clarity.
  2. Figure 4 has been modified, in order to improve the clarity.
  3. The analysis of the findings in figure 5 and 6 has been revised and modified.
  4. Additional literature has been cited to provide theoretical support for the analysis.

Furthermore, minor editing has been done to avoid typos and grammar mistakes.

Thanks a lot and best regards.

Reviewer 2 Report

Comments and Suggestions for Authors

The paper calls "Contacting of bicomponent TPU-fibres with a conductive core: A method for data acquisition and analysis of the electrical properties" and concerned of filaments laser cutting method. The advantage of article is actual topic of research and relate references list.

However reviewer have some questions:

1. What type of laser (gas, fiber or solid-state) do you use?

2. What is wavelength of operation of laser?

Article formatting is slid and some "references not found"...

Author Response

Dear Reviewer,

thank you for taking your time to review our paper. Please find attached our responses to your feedback:

  1. The type of laser used (CO2) is described in table 2.
  2. The wavelength (655 nm) has been added to table 2).
  3. Broken references and formatting has been fixed.

Thanks a lot and best regards

Reviewer 3 Report

Comments and Suggestions for Authors

In their work, the authors develop a cutting and contacting method for TPU core-conductive bicomponent filaments. Overall, the paper is well written, easy to read, interesting and provide useful information. Furthermore, some grammatical mistakes (singular, plural, etc.) and typos (incorrect/missing punctuation, sentences attached to each other, etc.) need to be corrected.

Below are a few points for the authors to consider to improve the quality of the manuscript.

 1-     Abstract. It should contain a summary of the results of the authors investigation. It should contain as well the variable range of different parameters

2-     The number of references is 13. This is too little and many of them are not accessible.

3-     How many sample did the authors conduct in their study?

4-     Line 67: “Error! Reference source not found.” Refer to what?

5-     Line 119: “Error! Reference source not found.” Refer to what?

6-     Line 147: “Error! Reference source not found.” Refer to what?

7-     Equation 1 must be retyped using Mathtype software.

8-     Line 199: “Error! Reference source not found.” Refer to what?

9-     Line 201: “Error! Reference source not found.” Refer to what?

10- Line 202: “Error! Reference source not found.” Refer to what?

11- Line 220-223-217: “Error! Reference source not found.” Refer to what?

12- Figure 7. Two decimal is enough for the regression equation and coefficient of determination.

13- Figure 8. The linear resistivity of each sample be added to the figure or as a separate table.

14- The conclusion section is very long. It should be summarized in a bullet form.

15- The authors should clearly mention the limitation of their study.

16- A separate recommendation section is needed.

Comments on the Quality of English Language

some grammatical mistakes (singular, plural, etc.) and typos (incorrect/missing punctuation, sentences attached to each other, etc.) need to be corrected.

Author Response

Dear Reviewer,

thank you for taking your time to review our paper. Please find attached our responses to your feedback.

  1. A summary of the results has been added to the abstract.
  2. All references are available to the authors. A further reference to provide theoretical support for the analysis has been added.
  3. Five samples have been analyzed for each cutting method. The number of samples has been added to chapter 2.2.2.
  4. Reference has been fixed.
  5. Reference has been fixed.
  6. Reference has been fixed.
  7. The equation has been typed according to the journals guidelines.
  8. Reference has been fixed.
  9. Reference has been fixed.
  10. Reference has been fixed.
  11. Reference has been fixed.
  12. The number of decimals has been reduced to two.
  13. Due to the high number of measurements taken (15 for each of the 8 samples), the data was not added as a table in the paper. The data however can be attached as supplementary material and is attached to this reply.
  14. A summary in bullet points has been added.
  15. A chapter on the limitations of the study has been added.
  16. For the authors it is not clear what is meant by a separate recommendation section. Our assumption is, that this refers to the outlook. The outlook has been added to the conclusion.

Furthermore, typos and and grammarly mistakes have been corrected.

Thanks a lot and best regards.

Reviewer 4 Report

Comments and Suggestions for Authors

This study employs laser cutting to expose the conductive core, followed by silver coating and crimping for both conductivity and protection and explores laser parameters that ensure a robust and reproducible method for strain-sensing applications with TPU filaments. While the work is interesting from the reviewer's standpoint, there are some concerns that the authors should address.

1.         In the discussion section, while the author provides reasonable explanations for qualitative characteristics, such as the striped visual effect observed during laser cutting at different speeds and powers, these interpretations seem to be largely speculative and based on the author's experience. Therefore, the reviewer suggests conducting additional comparative experiments to further investigate each characteristic and enhance the scientific rigor of the study.

2.         Some reference sources of this paper are not correctly linked.

3.         The explanation of the razor-cutting method appears overly brief. In addition to laser cutting, it would be beneficial to explore methods of razor cutting that avoid smearing over the conducting core. The reviewer suggests conducting more rigorous experiments to test the effectiveness of razor cutting and to propose improved cutting methods or demonstrate the necessity of laser cutting over razor cutting for this material.

Author Response

Dear Reviewer,

thank you for taking your time to review our paper. Please find attached our responses to your feedback:

  1. Sources were added, where the different morphologies of the cut surface are discussed. Further analysis of this topic exceeds the scope of this work. Instead, the electrical measurements show, that the laser parameters (and with that the morphologies) are inconsequential to the linear resistivity. Additionally, a chapter on the limitations of the study has been added.
  2. All references have been fixed.
  3. Different razor cutting methods have been investigated before this study. A description on previous trials has been added to chapter 3.1.

Thanks a lot and best regards.

Round 2

Reviewer 3 Report

Comments and Suggestions for Authors

The authors significantly improved their manuscript and properly addressed my comments. No further issues